# A multi-marker association method for genome-wide association studies without the need for population structure correction

Jonas R. Klasen[1,2], Elke Barbez[3], Lukas Meier[4], Nicolai Meinshausen[4], Peter Bühlmann[4], Maarten Koornneef[2], Wolfgang Busch[3] & Korbinian Schneeberger[1]

All common genome-wide association (GWA) methods rely on population structure correction, to avoid false genotype-to-phenotype associations. However, population structure correction is a stringent penalization, which also impedes identification of real associations. Using recent statistical advances, we developed a new GWA method, called Quantitative Trait Cluster Association Test (QTCAT), enabling simultaneous multi-marker associations while considering correlations between markers. With this, QTCAT overcomes the need for population structure correction and also reflects the polygenic nature of complex traits better than single-marker methods. Using simulated data, we show that QTCAT clearly outperforms linear mixed model approaches. Moreover, using QTCAT to reanalyse public human, mouse and Arabidopsis GWA data revealed nearly all known and some previously undetected associations. Following up on the most significant novel association in the Arabidopsis data allowed us to identify a so far unknown component of root growth.

[1] Department of Plant Developmental Biology, Max Planck Institute for Plant Breeding Research (MPIPZ), Carl-von-Linné-Weg 10, 50829 Cologne, Germany. [2] Department of Plant Breeding and Genetics, Max Planck Institute for Plant Breeding Research (MPIPZ), Carl-von-Linné-Weg 10, 50829 Cologne, Germany. [3] Gregor Mendel Institute (GMI), Austrian Academy of Sciences, Vienna Biocenter (VBC), Dr Bohr-Gasse 3, 1030 Vienna, Austria. [4] Seminar for Statistics, Department of Mathematics, Eidgenssische Technische Hochschule Zurich (ETHZ), Rämistrasse 101, 8092 Zurich, Switzerland. Correspondence and requests for materials should be addressed to J.R.K. (email: klasen@mpipz.mpg.de) or to K.S. (email: schneeberger@mpipz.mpg.de).

Since the advent of high-throughput genotyping methods, genome-wide association (GWA) is the emerging tool for studying the genetics underlying natural phenotypic variation. GWA studies have been applied across a wide range of species where it enabled fine-scale genetic mapping[1]. However, the individuals used in GWA studies are typically not evenly related to each other, but their relationships are influenced by population stratification and cryptic relatedness, here jointly referred to as population structure[2]. Population structure leads to linkage disequilibrium (LD) between physically unlinked regions and thereby to correlations between markers of these regions. When testing for the contribution of a particular marker on a phenotype, deviations from the expected independence between marker and phenotype are tested. As a consequence of population structure, this test can yield false associations between the phenotype and markers that are only correlated with, but not physically linked to causal variants[3]. In addition to population structure, the polygenic inheritance of complex traits further complicates the association of single markers to the phenotype[4], as true associations of causal variants and the phenotype can be masked by the genetic background, which summarizes the effects of all other loci in the genome.

Presumably the most widely used method for GWA is the linear mixed model (LMM), which tests the association of the phenotypes to each individual marker, while taking into account a so-called random effect[5–11]. This random effect corrects for population structure and the genetic background simultaneously. It models the genetic covariance between the individuals of the population using a similarity matrix, called genomic relationship matrix (GRM)[12], which is typically directly calculated from the markers. During the random-effect estimation, every marker in the GRM contributes a small fraction to the covariance of the individuals, assuming that many loci with small effects throughout the entire genome are contributing to the genetic background (referred to as infinitesimal genetic background).

However, the use of a random effect to correct for population structure and for an infinitesimal genetic background changes the hypothesis tested. Instead of testing whether a particular locus has an effect on the phenotype, it tests whether a locus has an effect on the phenotype that is neither explained by population structure nor by the genetic background. This is a drastic restriction to the hypothesis, which eventually leads to failures in the identification of causal loci[13,14].

To minimize these negative effects of population structure correction several improvements have been suggested, including relaxation of the (in many cases) unrealistic assumption of an infinitesimal genetic background by a more explicit modelling of the underlying genetics[15–18] or by correcting for population structure with other techniques than the random effect[19,20]. However, as population structure and genetic background correction are addressed simultaneously in the LMM approaches, improvements in handling of one of them can have negative consequences for the other. Therefore, improvements in population structure correction, such as calculating the GRM from a subset of markers, can still result in an impaired error control of the final associations[14].

Here we introduce a new GWA method, called Quantitative Trait Cluster Association Test (QTCAT) that accounts for the correlation between the markers, while associating them to the phenotypes and thereby avoids the problems introduced by population structure (that is, correlated, but physically unlinked markers). In addition, QTCAT also avoids the need to correct for any genetic background, as it simultaneously associates all markers to the phenotype. Using simulations based on populations with high levels of population structure, we showed that QTCAT is superior to LMM approaches, finding more loci with small effects, more extreme allele frequencies and higher involvement in population structure. Moreover, using simulations of experimental mapping populations without any population structure showed that QTCAT is also superior to common linkage mapping. To evaluate the performance of QTCAT on real data, we reanalysed previously published GWA data from Arabidopsis, mouse and human populations. Although nearly all previously reported associations were identified, QTCAT found additional associations in all cases. The most significant novel association found in an Arabidopsis data set for variation in root meristem zone lengths revealed an association of two candidate genes. The loss of function for one of the two genes resulted in a reduction of meristem zone length and therefore exemplified the great potential of candidate gene identification by QTCAT.

## Results

**How QTCAT works.** The polygenic nature of quantitative traits makes multi-marker association methods superior to single-marker associations, as they combine all markers into one model to search for the subset of markers that together explains the phenotype best[15,21]. This can be formulated as a linear model

$$\mathbf{y} = \mathbf{X}\boldsymbol{\beta} + \epsilon,$$

where the continuous response variable $\mathbf{y}$ is a vector of $n$ phenotypic observation, $\mathbf{X}$ is an $n \times p$ design matrix of covariates (markers), $\boldsymbol{\beta}$ is a vector of $p$ effects and $\epsilon$ is a vector of $n$ residuals. $p$, the number of markers, is usually much larger than the number of individuals $n$, resulting in high-dimensional linear model where $p \gg n$. However, as usually only a few markers have an effect on the phenotype, the model is sparse with many elements of $\boldsymbol{\beta}$ being equal to zero.

This is a common model selection problem with the additional complication that the markers are correlated. As a consequence, highly correlated markers are exchangeable as members of the 'best subset', implying that there is not one unique subset of markers that explains the phenotype best. To address this, QTCAT searches for clusters of highly correlated markers that are significantly associated to the phenotype instead of searching for individual markers. For this, QTCAT follows an earlier suggestion by Meinshausen[22], who proposed a hierarchical testing procedure for correlated covariates. This concept starts by generating a hierarchical clustering of all covariates based on their correlations, followed by testing these clusters for significant associations to the response variable along this hierarchy. Testing starts at the top of the clustering tree and continues by testing the clusters of the next lower level of the clustering hierarchy. This is repeated for all those clusters that have been significantly associated until none of the clusters of the next lower level are significantly associated anymore or until the single covariates level is reached. The lowest, still significant clusters in the hierarchy are the final result clusters, which include all those covariates that are significantly associated to the response variable. Significance testing of the individual clusters is not only considering the covariates of the actual cluster, but contrasts their influence on the phenotype with the influence of all other covariates (for example, by using analysis of variance (ANOVA) tests), which guarantees that their influence is not overestimated.

This idea fits particularly well to the challenges of GWA. Each of the final clusters, called quantitative trait clusters (QTCs), combines markers, which cannot be distinguished for their individual contribution to the phenotypic variation. Important to note, this refers to markers, which are highly correlated to each other. Markers, which are not highly correlated, will not be able to explain the same part of the phenotypic variation and will not become part of the same QTC. Moreover, during the hierarchical

testing procedure, the significant association of a QTC will suppress potential associations of other clusters to the same part of the phenotypic variation, ensuring that a particular part of the phenotypic variation is only explained by one QTC. With this, false associations, including erroneous associations introduced by population structure, are effectively avoided even without any further correction terms (Fig. 1).

To make this method applicable to high-dimensional linear models ($p \gg n$) we implemented the hierarchical testing as a repeated sample-splitting procedure[23,24] into a new algorithm called Hierarchical Interference Testing (HIT). HIT starts by selecting a random subset of individuals to identify a set of informative markers using a LASSO[25] approach. This drastically reduces the number of markers, which then can be tested for significant associations to the phenotype along the clustering hierarchy as described above (using sequential ANOVAs as significance tests and only those individuals that were not used for the initial marker selection). For significance testing, only markers that were pre-selected by the LASSO are used. Sample splitting is repeated for multiple times, where each iteration starts with a different set of randomly selected individuals and therefore different sets of markers are tested in each repetition. Finally, the results of all iterations are summarized (Supplementary Note for more details).

**Comparison to LMMs using structured populations.** To compare the performances of QTCAT and LMM we simulated GWA data using the genotypes of 1,307 accessions of the Arabidopsis RegMap panel, which include high levels of population structure[26]. On top of these genotypes we simulated three different phenotypes each with 100 replicates. The first phenotype was based on 50 causal loci with effects that were randomly drawn from a Gamma distribution and heritability ($h^2$) of 0.7. The second phenotype was similar though the effects were drawn from a normal distribution instead, to avoid the effects of predominant major loci. The third phenotype was more complex and based on 150 loci with effects that were randomly drawn from a normal distribution and an $h^2$ of 0.4. We associated these phenotypes to the genotypes using QTCAT, as well as two different LMM approaches (Supplementary Figs 1–300). For the first LMM approach we used all markers for GRM estimation (LMM), whereas for the second approach we left out the markers on the same chromosome as the tested position for GRM

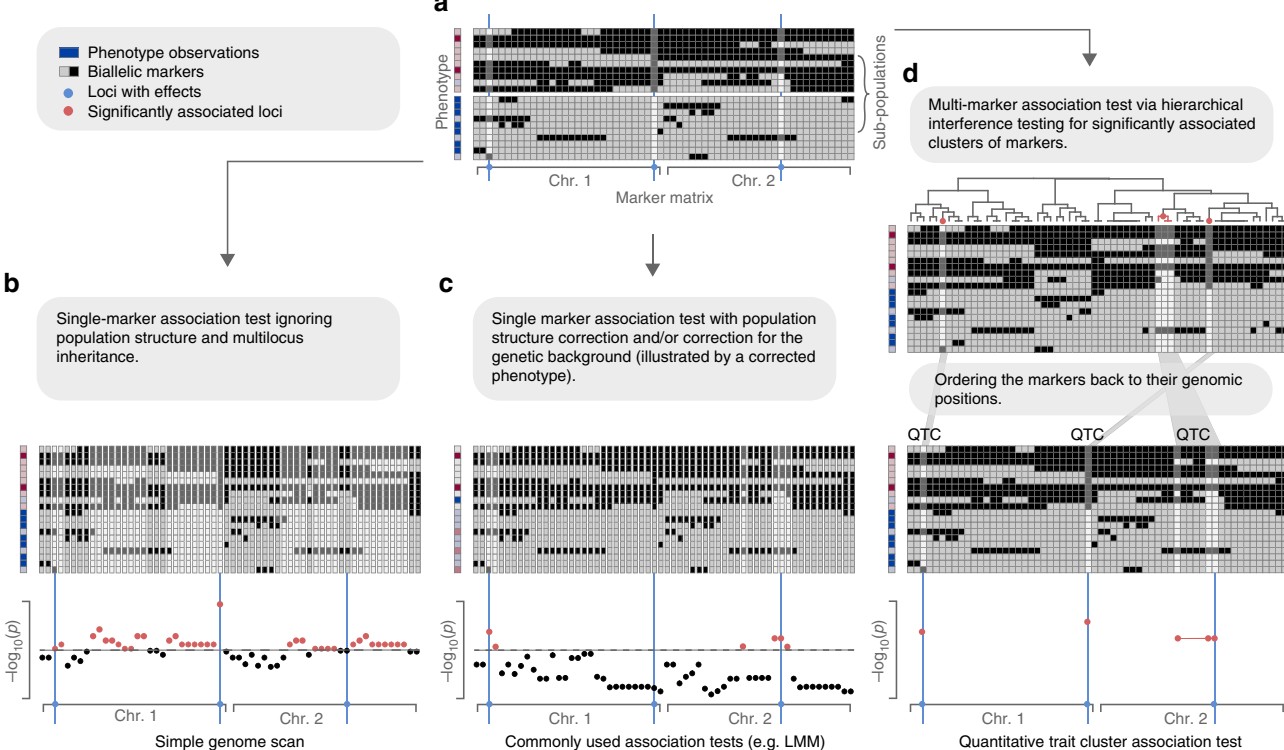

**Figure 1 | Schematic of QTCAT's workflow and other common methods.** (**a**) A marker matrix representing individual genotypes (rows) of a population with two alleles (light and dark gray) per marker (columns). The population consists of two distinct subpopulations, where the individuals of each subpopulation are highly related. The loci of the markers with effects on the phenotype (causal loci) are highlighted in blue. The phenotype is shown on the left; the phenotypic values rank from dark red (if all three causal loci carry the dark grey allele) to dark blue (if all the loci carry the light grey allele). (**b**) A simple genome scan testing for associations between the phenotype and individual makers without considering population structure or genetic background. This results in a large amount of false, but significant associations due to the correlation of the markers. (**c**) Associations tests between single markers and the phenotype, while controlling for population structure and genetic background (illustrated by a corrected phenotype). This results in a drastic reduction of false associations; however, also some of the real associations cannot be identified. (**d**) Simultaneous association of all markers against the phenotype, while considering the correlations between the markers. First, all markers are clustered according to their correlation. All possible clusters of markers (that is, all nodes of the clustering tree) are then hierarchically tested for their significant association to the phenotype. All smallest clusters that are significantly correlated to the phenotype are reported as QTCs. In addition, the significant association of a QTC will suppress the association of other clusters to the same part of the phenotypic variation and thereby avoids false associations, including those introduced by population structure. This allows associating markers and phenotype without the restrictive effects of population structure correction and corresponds well with the multi-genic inheritance of complex traits.

estimation (LMM*), which showed improved results compared with the first LMM approach in earlier studies[13,14]. As QTCAT's significance testing is implemented as family-wise error rate (FWER), we also applied FWER (Bonferroni) correction to the results of both LMM approaches.

Throughout all simulations we observed that QTCAT had a highly efficient error control reporting almost no false positives. Interestingly, these low error rates were even almost entirely insensitive to changes in the $\alpha$-levels (Fig. 2). According to this, QTCAT's precision—here defined as the percentage of true positives of all reported loci at an $\alpha$-level of 5%—was on average 98.4, 99.5 and 98.1% for the three different phenotypes. For the first two phenotypes the precision of the LMM approach was similar (on average 97.4 and 97.9%); however, the precision for the third and most complex phenotype was only 92.5% and thus considerably lower. This drop in precision of the LMM is likely to be due to the high genetic complexity in regard to sample size and might be overcome with more individuals. Moreover, the precision of LMM* was drastically lower throughout all simulations (on average 81.9, 88.5 and 86.8% for the three phenotypes) and was also even lower compared with the LMM, which is in contrast to previous reports on human data[13,14], but might be a consequence of the small number of chromosomes of Arabidopsis, implying that large parts of the genome are excluded during GRM calculation. Despite being less error prone, QTCAT identified more of the causal loci at any given $\alpha$-level for the first two phenotypes as compared with both LMM approaches (Fig. 2). For the third phenotype, QTCAT's stringent error control led to less true positive loci at an $\alpha$-level of 5%, but at the same level of false positives the number of true positives was similar.

To analyse these results in more detail, we explored the outcome of the associations to the first phenotype at an $\alpha$-level of 5%. As mentioned, QTCAT found significantly more loci compared with the results of LMM (on average 8.48 versus 6.71 loci; $P$-value = 4.094e–09, Wilcoxon test), whereas the average number of false positives was very low for both methods (on average 0.14 versus 0.18, $P$-value = 0.4391, Wilcoxon test; Fig. 3a). Changing the multiple testing correction of the LMM to the commonly used false discovery rate (Benjamini–Hochberg[27]) increased the average number of correctly identified loci to 8.52, which was very similar to the results of QTCAT (8.48 versus 8.52, $P$-value = 0.9477, Wilcoxon test). However, at the same time the number of false positives was drastically increased to 3.93 as well. This was not only more as compared with the number of false positives of QTCAT (0.14 versus 3.93,

$P$-value < 2.2e − 16, Wilcoxon test), but accounts for nearly one third of all loci found by the LMM. Similar to that for LMM, the average number of true positives estimated by LMM* was significantly lower as compared with QTCAT (8.48 versus 7.17, $P$-value = 5.011e − 06, Wilcoxon test), whereas the average number of false positives was already significantly increased when using FWER for multiple testing correction (0.14 versus 1.59, $P$-value = 1.538e − 11, Wilcoxon test).

Comparing QTCAT with the LMM, 76% of the loci were identified by both methods, whereas 22% were found exclusively by QTCAT and only 2% were found by the LMM alone. Consequently, the results of QTCAT explained significantly more of the variance than the results of the LMM (0.62 versus 0.57, $P$-value = 3.82e − 10, Wilcoxon test; Fig. 3b). Although large effect loci were found by both methods equally well, QTCAT could expand its findings to loci with smaller effect size (Fig. 3c). The majority of the loci identified by QTCAT alone had significantly lower minor allele frequencies than the loci identified by both methods (on average 0.26 versus 0.30, $P$-value = 0.02765, Wilcoxon test; although considering only loci with similar effects (between 1.1 to 1.6) to be independent of effect size; Fig. 3d) and, consequently, the average variance explained by these loci was significantly lower as well (0.03 versus 0.06, $P$-value = 8.697 e − 10, Wilcoxon test; Fig. 3e). Intriguingly, population structure explained on average 12% of the variance of the markers found by both methods and 18% of those found by QTCAT alone ($P$-value = 0.005335, Wilcoxon test; again only considering loci with similar effect sizes).

Together, this suggests that despite making less errors QTCAT identifies more loci, which are typically more difficult to find, including those with low effects sizes and low minor allele frequencies, as well as loci with a much higher involvement in population structure (Fig. 3f and Supplementary Methods).

**Comparison to MQM using unstructured populations..** Even though QTCAT can be applied to populations with population structure, it does not require population structure and can also be applied to populations without any population structure at all. To test this, we compared the performance of QTCAT and MQM[21], which is a commonly used multi-marker linkage mapping method. The comparison was based on simulated biparental mapping populations with 1,307 recombinant inbred lines derived from 2 diverse Arabidopsis accessions. Again, we simulated a phenotype with 100 replicates with an $h^2$ of 0.7

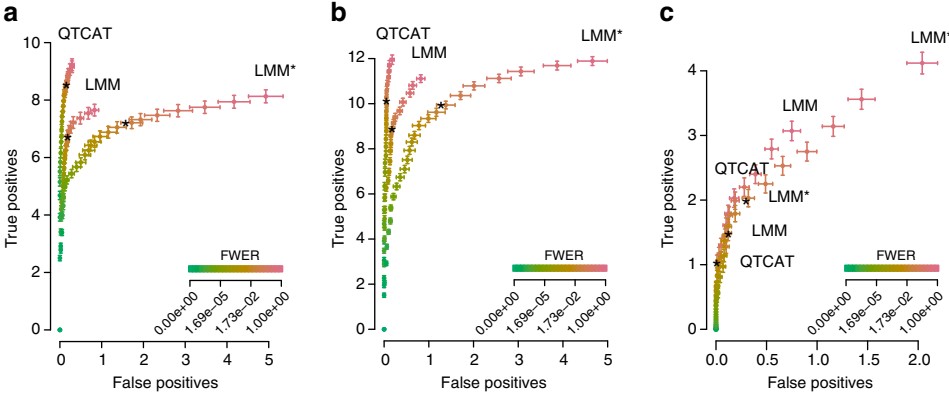

**Figure 2 | True-positive versus false-positive loci at different $\alpha$-levels for three different association methods.** Each plot is generated on the basis of simulations with 100 replicates (see main text). The error bars represent s.e. The stars indicate the amount of true and false positives at an alpha level of 0.05. (**a**) Phenotype 1: results for simulations based on 50 causal loci with effects that were randomly drawn from a Gamma distribution and a heritability ($h^2$) of 0.7. (**b**) Phenotype 2: similar simulation study as in **a**; however, effects were drawn from a normal distribution. (**c**) Phenotype 3: results for simulations based on 150 causal loci with effects that were randomly drawn from a normal distribution and an $h^2$ of 0.4.

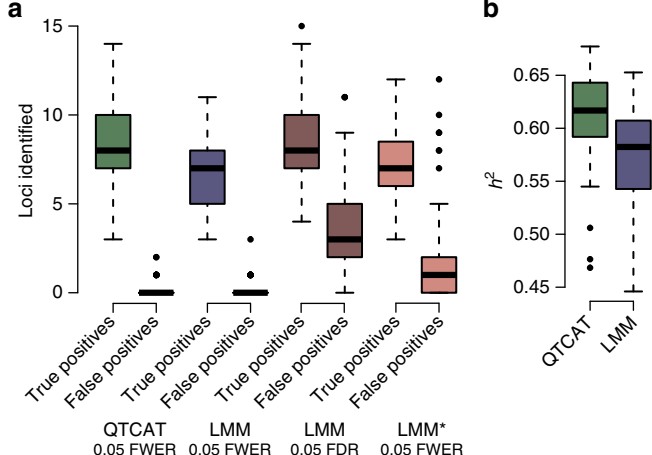

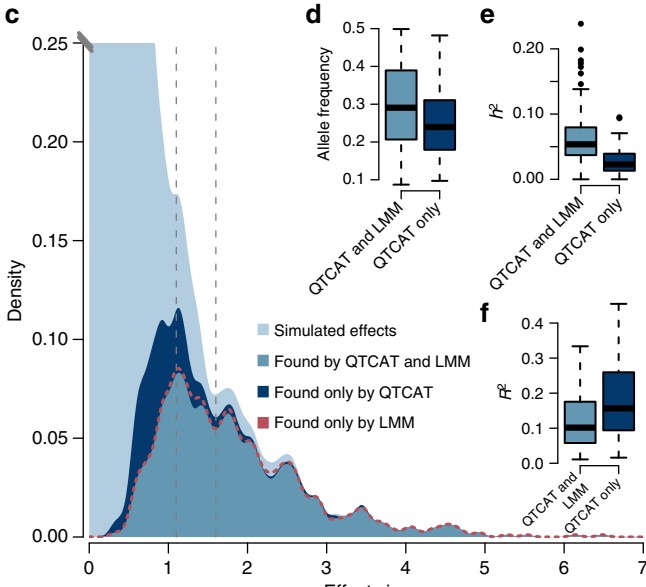

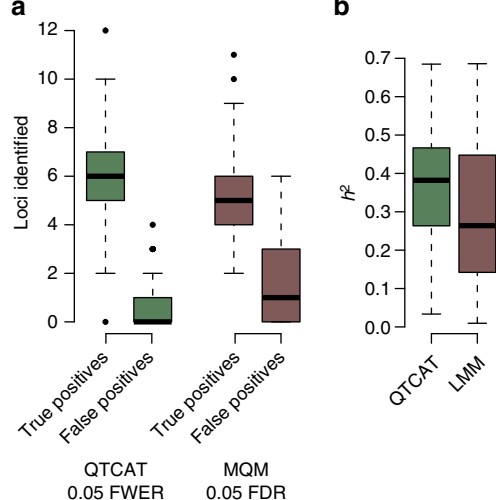

**Figure 4 | Linkage mapping results based on simulated biparental populations.** (**a**) Differences in the number of true and false positive loci identified by QTCAT and MQM (P-value = 0.006721, Wilcoxon test). (**b**) Differences in the explained variances of the loci identified by QTCAT and MQM (P-value = 0.01336, Wilcoxon test). Boxplots show median (bold line), the first to third quartile (box) and whiskers showing span of remaining data (max. 1.5 times of the box size).

QTCAT identified significantly more causal loci (on average 5.98 versus 5.35, P-value = 0.006721, Wilcoxon test) and significantly less false positives (0.71 versus 1.62, P-value = 5.033 e − 07, Wilcoxon test; Fig. 4a). The results of QTCAT and MQM explained on average 0.36 and 0.31 of the variance (P-value = 0.01336, Wilcoxon test; Fig. 4b). Though biparental populations have no population structure and an allele frequency of ∼0.5, they comprise extended regions with strong linkage due to the low number of recombination events introduced during their generation. These long haplotypes can lead to 'ghost quantitative trait loci (QTLs)'[28], which occur in between two causal linked loci as the low resolution hampers the separation of both signals. Across all 100 simulations we found 17 'ghost QTLs' predicted by MQM. In all cases, QTCAT did not report the 'ghost QTL' but either one or both of the neighbouring causal loci. Together, this suggests that proper consideration of correlated markers can also be beneficial for the analysis of biparental mapping populations.

**QTCAT revealed an unknown root growth factor of Arabidopsis.** In a recent GWA study, Meijón et al.[29] have used a world-wide collection of 201 natural Arabidopsis accessions to study the genetic basis of root development. Using an LMM association method with an FWER of 0.1, they identified one significant association, which subsequently revealed an F-box gene, in which natural genetic variation influences the meristem zone lengths in roots. We reanalysed these data with QTCAT and found four significant loci at an FWER of 0.05, including the previously reported locus as the most significant association (Fig. 5a,b). The second most significant QTC contained three closely linked markers located in two neighbouring genes (Fig. 5c). One of them, *PEPR2*, codes for a leucine-rich repeat receptor kinase and was previously shown to be a receptor for endogenous peptides enhancing innate immunity in roots[30]. One of the markers altered the encoded protein sequence by changing a glycine residue to a serine in the LRR domain of the *PEPR2* gene (Supplementary Fig. 401).

To test whether the *PEPR2* gene activity influences root meristem size and to provide evidence for the allelic contribution

**Figure 3 | Analysis of the results of GWA simulations using structured populations (Phenotype 1).** (**a**) Number of true and false positive loci identified by QTCAT and LMM (with multiple testing correction based on FWER and false discovery rate (FDR) shown separately). (**b**) The explained variance of the loci identified by QTCAT and LMM within the 100 simulations. (**c**) The distribution of all simulated effects (light blue) and the distribution of effects of loci identified by QTCAT (dark blue) and the intersection of found loci of QTCAT and LMM (blue). The dashed red line shows the findings of the LMM. Although QTCAT and LMM find almost all of the large effect loci, QTCAT is more efficient in finding small effect loci. (**d**) Difference between the allele frequencies of loci found by both methods and by QTCAT alone (P-value = 0.02765, Wilcoxon test). (**e**) Differences between explained variances of loci found by both methods and by QTCAT alone (P-value = 8.697e − 10, Wilcoxon test). (**f**) Compared with the loci identified by both methods, loci found by QTCAT alone were significantly more involved into population structure (P-value = 0.005335, Wilcoxon test one-sided). Boxplots show median (bold line), the first to third quartile (box) and whiskers showing span of remaining data (max. 1.5 times of the box size). Population structure involvement was estimated by the explained variance of the first five principal coordinates of the GRM regressed to the individual markers. It is noteworthy that for **d**, **e** and **f**, only loci with similar effect sizes were considered (between 1.1 and 1.6, see grey dashed lines in **c**).

and 20 causal loci with effects that were randomly drawn from a Gamma distribution (Supplementary Methods and Supplementary Figs 301–400).

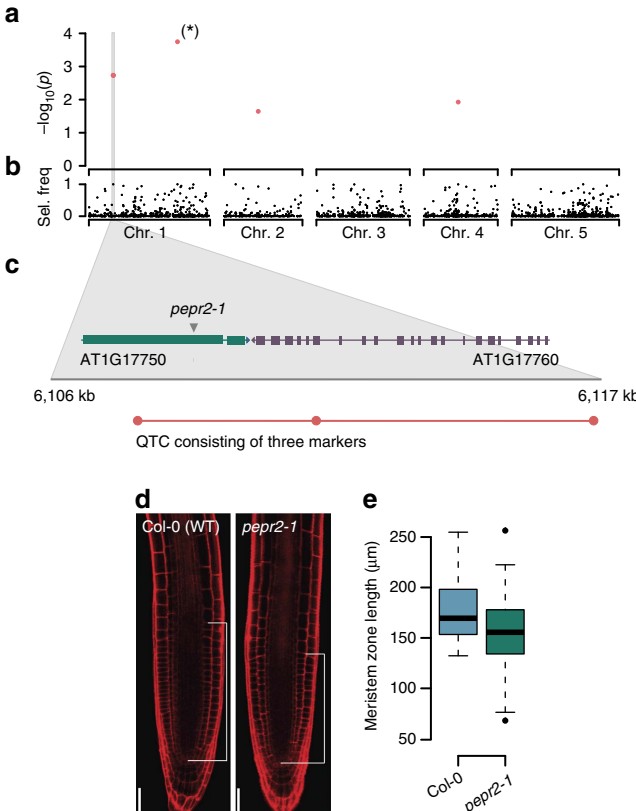

**Figure 5 | GWA (QTCAT) analysis and validation of an Arabidopsis population including 201 accessions phenotyped for differences in root meristem zone length.** (**a**) Typical output of QTCAT showing four significant QTCs for variation in root meristem zone length. The QTC marked with a star refers to the association, which was also reported by Meijón et al.[29]. (**b**) Selection frequency of markers during 500 repeated sample splittings of QTCAT. (**c**) Zoom-in on the QTC on the upper arm of chromosome 1 including all three markers of this QTC. The QTC overlaps with two candidate genes, including *PEPR2*, which encodes a leucine-rich repeat receptor kinase expressed in the roots. (**d**) The root middle sections of three days-old Col-0 and *pepr2-1* mutant seedlings. White bars indicate meristem zone length (scale bar, 50 μm). (**e**) Boxplots of meristem zone lengths of Col-0 and *pepr2-1* (four independent experiments with 57 Col-0 and 67 *pepr2-1* plants in total; *P*-value = 0.0007, ANOVA). Boxplots show median (bold line), the first to third quartile (box) and whiskers showing span of remaining data (max. 1.5 times of the box size).

of this gene to the observed natural variation, we compared the meristem length of wild-type Col-0 and *pepr2-1* loss-of-function mutant seedlings under the same conditions as described in Meijón et al.[29]. *pepr2-1* seedlings displayed significantly shorter meristems compared with the wild type (*P*-value = 0.0007, ANOVA; Fig. 4d,e). Interestingly, this effect was specific to the growth conditions that were used to conduct the original study as a prolonged stratification of the seeds abolished the effect (*P*-value = 0.1992, ANOVA; Supplementary Fig. 402). Even though it remains to be seen whether the impact of *PEPR2* on meristem size is due to processes relating to germination or root meristem size control, these data clearly demonstrate that the new association identified by QTCAT results from causal genetic variation, which has been missed with the earlier LMM approach.

**Applying QTCAT to mouse and human data sets**. We further applied QTCAT to 90 mouse genotypes from the Hybrid Mouse Diversity Panel[31], which have been phenotyped for high-density lipoprotein levels. In contrast to the Arabidopsis population

consisting of natural accessions only, this population combines classic inbred and recombinants of such strains, which strongly increases linkage and therefore the correlations between markers. Even though removal of perfectly correlated markers is a common step in QTCAT, it has a particularly large effect in populations with a limited number of recombination events. Here, QTCAT started by clustering 1,203,594 markers but only 73,091 remained in the clustering, as all other markers were perfectly correlated to at least one of the remaining markers. Removing redundant markers avoids unnecessary complexity during association without any disadvantage, as they are reintroduced into the final QTCs and thereby contributes to the final localization of the QTCs. A previous GWA based on an LMM approach revealed three significant loci within this data: one dominant peak at the bottom of chromosome 1, one on chromosome 11 and one on chromosome 15 (ref. 31). QTCAT recovered the dominant locus on chromosome 1, whereas the association on chromosome 15 was untangled into two distinct QTCs. Even though QTCAT could not retrieve the peak on chromosome 11, it did report on two additional QTCs on chromosome 5, which previously had not been identified (Supplementary Fig. 403).

We further applied QTCAT to human case-control data of the Wellcome Trust Case-Control Consortium[32]. In contrast to LMM approaches, which assume normally distributed phenotypes, QTCAT can model binomially distributed phenotypes, which is in better agreement with the actual case–control data. The Wellcome Trust Case-Control Consortium data comprise 11,341 cases (multiple sclerosis patients) and two control cohorts with 2,930 controls from the 1958 British Birth Cohort and 2,737 controls from the National Blood Donors Cohort. Markers were filtered to a high quality set of 304,638 markers, which we analysed using QTCAT and an LMM for direct model comparison, as this data set was originally combined with other data[32] (Supplementary Fig. 404 and Supplementary Methods). The results of the LMM were almost a perfect subset of the results of QTCAT, as 12 loci had been identified by both methods (using a FWER of 5%) but only one locus was identified by the LMM alone, whereas seven loci were found exclusively by QTCAT. Considering the low error rates of QTCAT, this result promises to reveal genes that so far could not have been identified and more generally again shows the great potential of QTCAT including its applicability to human case–control data.

## Discussion
Over the recent years, several improvements to the LMM-based association approach have been proposed[15–20]. Despite these advancements, all improvements are based on the assumption that population structure correction along with its negative effects cannot be entirely avoided, in particular if the trait is not approximately following an infinitesimal genetic architecture. Here we showed that testing of markers with a high-dimensional variable selection procedure, which can account for the correlations between the markers, does not require any population structure correction at all.

The basic concept of QTCAT is to combine those highly correlated markers, which cannot be distinguished for their individual contribution to the phenotype (Supplementary Fig. 405). In most cases, this refers to markers that are physically very close to each other. If this is not the case and physically unlinked regions are highly correlated (for example, by co-selection of physically unlinked regions), QTCAT will report the markers of both regions within one QTC. Although, in these rare cases, there would not be a unique association to one region (but to two regions), alternative approaches would treat

both regions independently, missing out on the fact that they cannot be distinguished for their contribution to the phenotype anyways.

Population structure is not species specific, but can be found in populations of any type. QTCAT is entirely independent of the system, including mating types and heterozygosity levels, and can also consider additional covariates to control for sex differences similar to the LMM approach. Only in cases where the causal variants are not closely physically linked to any of the markers, incorrect regions might be reported as QTCs. However, such scenarios are more and more unlikely, as increasing application of high-throughput whole-genome sequencing methods for genotyping ensures dense marker sets.

Non-genetic factors can also influence the phenotype (if not controlled for in the study design). Even though there is no causal connection between phenotype and genotype in these cases, this can result in false positive associations. If non-genetic factors co-occur with population structure, population structure correction can avoid these false associations. As QTCAT does not rely on population structure correction, such factors need to be explicitly modeled to guarantee stringent error control.

All common genetic mapping methods include two main steps: first, detection of associations and, second, precise localization of a mapping interval around each association that has a high probability of containing the causal variant[33]. Conventional linkage mapping uses genetic maps, estimated from the amount of recombination in the population, to define such mapping intervals. As genetic maps cannot be calculated from natural populations, LD decay and haplotypes are usually used to approximate mapping intervals in such populations. QTCAT is fundamentally different in this regard: both steps are performed simultaneously as the markers of a QTC account for both the association and its location. When applied to biparental mapping populations, QTCs act as an equivalent to mapping intervals of linkage mapping approaches. In natural populations, QTCs are found around causal variants representing the extent of LD within this region. In the simulation based on Arabidopsis populations, only 6.6% of all QTCs were >10 kb (0.6% >50 kb), which agrees well with the average LD decay in Arabidopsis[34].

Statistical solutions for high-dimensional regression based on correlated covariates are still in their infancy and further improvements are likely to follow soon. Our simulations showed that QTCAT is still very conservative, indicating that such further development could lead to even more powerful methods. However, already in the current form QTCAT properly accounted for polygenic inheritance and helped overcoming the need for population structure correction. In any case, independent of the actual method, associating clusters of highly correlated markers (QTCs) will always be superior to single-marker association, as they are more consistent with the nature of quantitative traits.

## Methods

**Simulation of structured populations.** GWA data were simulated from a set of 214,051 single-nucleotide polymorphism markers, which were genotyped for 1,307 diverse Arabidopsis accessions showing strong population structure[26]. On top of this genetic data, we simulated three different phenotypes with 100 replicates for each phenotype. In each replicate of the first phenotype 50 markers were randomly selected as causal loci considering gene density, to guide causal loci to gene-dense regions. We assigned an additive effect randomly drawn from a $\gamma$-distribution[35] (shape = 0.5, scale = 1) to a random allele of each of these markers. In addition, we added a random environmental term so that $h^2$ of the simulated traits was 0.7. The second set of phenotypes was similar with the exception that the effects were drawn from a normal distribution. Finally, the third phenotype was based on 150 causal loci with effects that were randomly drawn from a normal distribution with a $h^2$ of 0.4.

**Simulation of unstructured populations.** For linkage mapping we simulated a recombinant inbred line (RIL) population using the genotypes of two diverse

Arabidopsis accessions (Col-0 and Ler)[26]. Recombination were randomly distributed over the genome following empirically assessed recombination frequencies and distributions[36]. The phenotypes were simulated similar as for the structured populations, with the difference that only 20 markers were assigned with an effect.

**Statistical methods.** All LMM analyses were performed with the R package rrBLUP[37], except the human data analysis where for performance reasons GRAMMA-gamma[10] was used. GRM was in general estimated from all markers, but for LMM* model five GRMs were calculated for each leaving one chromosome out. Two different error controls were used, a 5% FWER (Bonferroni) and a 5% FWR (Benjamini–Hochberg). To determine whether causal markers were correctly identified, a 10 kb region around each significant peak was used as final interval. For linkage mapping we used the MQM implementation in the R package R/QTL[38]. MQM was used to detect QTLs with a 5% false discovery rate (simulation based permutation test) and localization was performed using the lodint-function of R/QTL, which was extended to allow for multiple QTLs per chromosome. QTCAT was run with 50 sample splittings for the simulated data sets and 500 for the Arabidopsis and mouse data. LASSO was used with tenfold cross-validation. FWER of QTCAT was set to 5%. The assumptions of normal distribution and homogeneous variances were validated by a regression of the QTC medoids and the phenotypes in all cases. Comparisons of the findings of QTCAT with the LMM and MQM results were performed by a Wilcoxon test. The degree of population structure involvement per marker was estimated by the explained variance of the first five principal coordinates of the GRM regressed to the individual markers.

**Computational time of QTCAT.** The simulations were performed on a computer with 2.3 GHz processors (AMD Opteron) running a Linux 64 bit operating system. Clustering of 214,051 markers and 1,307 individuals took ~17 h using QTCAT's clustering algorithm running in parallel on four cores (the cluster hierarchy might be pre-given and this computational step would then not be present). The actual association step performed by QTCAT's HIT algorithm took ~1 h and 45 min, also running in parallel on four cores. However, QTCAT could easily be run on many more parallel cores, as the algorithm is tailored for direct distributed computing to achieve substantial computational speed-up.

**Measuring and analysing Arabidopsis meristem zone length.** Arabidopsis seeds of Col-0 and pepr2-1 (ref. 30) genotypes were sterilized using 70% ethanol, air dried and subsequently seeded on $1 \times$ MS in vitro growth plates, pH 5.7, containing 1% (w/v) sucrose and 0.8% (w/v) agar. The seeds were stratified for 2 or 5 days, respectively, at 4 °C in the dark and subsequently germinated upright in long-day conditions (16 h light/8 h dark) at 21 °C. Three/six days old seedlings were mounted on microscopy slides in liquid growth medium supplemented with 20 μM propidium iodide (Sigma Aldrich) and analysed using a Zeiss 700 inverted confocal microscope equipped with a $\times 20$ air objective. Image identities were randomized before meristem measurements using the Fiji software (http://fiji.sc/Fiji) as described previously[39]. Resulting data of four independent experiments with 57 Col-0 and 67 pepr2-1 plants were analysed with ANOVA F-tests, while accounting for the different experiments. Normal distribution of data and homogenous variance was given.

**Data availability.** QTCAT is available as open source R package at http://github.com/QTCAT/qtcat/.

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

## Acknowledgements

We thank Yusuke Saijo for kindly providing *pepr2-1* seeds, and Jose M. Jimenez-Gomez, Pádraic J. Flood and George Coupland for their comments on the manuscript. This work has been generously supported by the Max Planck Society. This study makes use of data generated by the Wellcome Trust Case-Control Consortium. A full list of the investigators who contributed to the generation of the data is available from www.wtccc.org.uk. Funding for the project was provided by the Wellcome Trust under award 076113, 085475 and 090355.

## Author contributions

J.R.K. and K.S. designed the study. J.R.K. designed and implemented QTCAT. E.B. and W.B. performed the root growth experiments. J.R.K. performed data analyses. L.M., N.M. and P.B. supervised statistical aspects of this work. M.K. supervised biological aspects of this work. J.R.K. and K.S. wrote the manuscript with contributions from all other authors.

## Additional information

**Competing financial interests:** The authors declare no competing financial interests.

