## [Peer Review File · Nature Communications]

Reviewers' comments:

Reviewer #1 (Remarks to the Author):

Thank authors for addressing my comments on your second revision submitted to Nature Genetics. I still have concerns regardless publication elsewhere:

1. Concern on previous comment 1: I am glad to see that the proposed method had advantage over LMM on a setting with 50 genes with even effect. This is a good sign.
2. Concern on previous comment 2: A high heritable trait does not mean it is lack of influence from a big non genetic effect. For example, when a big effect is added to an ethnic group, the trait will have bi-mode distribution. However, the heritability stays the same after the correction of the fixed non genetic effect. My concern is how the proposed method would perform in this situation.
3. Concern on previous comment 3 & 4: The proposed method should have at least one advantage, either on computing speed, or statistical power, compared with existing methods. The manuscript made comparison with LMM that is at least ten-year-old technology. There are much more developments since then. It is OK that the authors choose not compare with MLMM on computing time. In this case, it is necessary to see if the proposed method gain anything on statistical power, compared with some new developments, such as FarmCPU.

Reviewer #2 (Remarks to the Author):

The work at hand presents a significant methodological advancement to GWAS that shows significant benefits in a wide range of experimental settings and genetic architectures.

The manuscript is clear and the reviewers questions and comments have been well addressed.

REVIEWERS' COMMENTS:

Reviewer #1 (Remarks to the Author):

Thank you for addressing my concerns. The manuscript is ready for publication.

Reviewers' comments:

Reviewer #1 (Remarks to the Author):

Thank authors for addressing my comments on your second revision submitted to Nature Genetics. I still have concerns regardless publication elsewhere:

1. Concern on previous comment 1: I am glad to see that the proposed method had advantage over LMM on a setting with 50 genes with even effect. This is a good sign.

2. Concern on previous comment 2: A high heritable trait does not mean it is lack of influence from a big non genetic effect. For example, when a big effect is added to an ethnic group, the trait will have bi-mode distribution. However, the heritability stays the same after the correction of the fixed non genetic effect. My concern is how the proposed method would perform in this situation.

The correction for such non-genetic effects in GWA studies should be similar as for the heritability estimation, regardless of the association method. As we have described in our first response QTCAT (like the LMMs) can account for additional covariates (fixed effects) and is thereby able to correct for big non-genetic effects.

To show how QTCAT performs given the presence of a big non-genetic effect we added a strong effect (approximately two times as large as the largest gene effect), to our first simulation scenario. This resulted in a bimodal distribution of the phenotype.

The results of associating these data with QTCAT and LMM are shown in the following plot. QTCAT again clearly shows very low levels of false negatives, while the recall rate is higher as compared to the LMM.

Moreover, in a second analysis, we simulated residuals, which were correlated to the population structure of the underlying genotypes. This mimics situations in which population structure and non-genetic effects co-occur, as we have discussed it in our first and second responses. Under these circumstances it is not obvious how to correct for the non-genetic effects, which is why the association methods themselves need to be robust against such influences. The LMM approach controls for inflated false positives due to such non-genetic effects with the random effect as part of the population structure correction. In contrast, QTCAT, which does not correct for population structure, accounts for the correlation of markers which makes it similarly robust against inflation of false positives.

In detail, during phenotype simulation, we simulated residuals, which were drawn from a multivariate normal distribution using the GRM as covariance matrix.

We again analyzed the data with a LMM and QTCAT. The results showed the robustness of both methods against such influences.

Taken together, both simulations underline how robust QTCAT is against the influences of non-genetic effects.

3. Concern on previous comment 3 & 4: The proposed method should have at least one advantage, either on computing speed, or statistical power, compared with existing methods. The manuscript made comparison with LMM that is at least ten-year-old technology. There are much more developments since then. It is OK that the authors choose not compare with MLMM

on computing time. In this case, it is necessary to see if the proposed method gain anything on statistical power, compared with some new developments, such as FarmCPU.

As we have described in our last response, methods which subset the GRM cannot guarantee the expected level of error control (Yang et al. 2014, Nat Genet). FarmCPU for example reported more than 50% false positives at a FWER of 0.05 in our third simulation scenario (see figure below).

In contrast QTCAT hardly reported any error, which is a clear advantage of QTCAT (despite finding less true positives), and which is based on a clearly defined theoretical foundation. As described in our manuscript, if the correlations between markers are correctly addressed, error control can be guaranteed.

Furthermore, we showed earlier that QTCAT has power advantages compared to the LMM approach at similar error control. Thus we strongly believe that this is a theoretical as well as a practical advantage compared to other methods. We now state this explicitly in the manuscript.

Reviewer #2 (Remarks to the Author):

The work at hand presents a significant methodological advancement to GWAS that shows significant benefits in a wide range of experimental settings and genetic architectures.

The manuscript is clear and the reviewers questions and comments have been well addressed.

Thank you, we appreciate this positive response!

REVIEWERS' COMMENTS:

Reviewer #1 (Remarks to the Author):

Thank you for addressing my concerns. The manuscript is ready for publication.

Thank you, we appreciate this positive response!